# Rare variants and founder effect in the Beauce region of Quebec
Mylène Gagnon[1,2], Claudia Moreau[1,2], Jasmin Ricard[3], Marie-Claude Boisvert[3], Alexandre Bureau[3,4], Michel Maziade[3,5] & Simon L. Girard [1,2,3,6] ✉

Founder events influenced the genetic diversity within the province of Quebec, increasing the frequency of certain rare pathogenic variants in regional populations. Some regions, such as Beauce, remain understudied despite evidence of a regional founder effect. Leveraging extensive genealogical data, we observe a specific regional structure emerging in Beauce following the initial settlement. It is characterised by a gradual increase in inbreeding and kinship coefficients and a low diversity of ancestors. Taking advantage of the region's genetic distinctiveness, we describe 36 rare pathogenic variants with higher carrier rates in Beauce than in urban regions, likely due to the regional founder effect. This provides the first in-depth study of Beauce's genetic and genealogical landscape, revealing a distinct structure and suggesting that other overlooked regions, in Quebec and elsewhere, could benefit from fine-scale population structure studies to improve the understanding and management of rare diseases.

The genetic specificity of populations with founder effects provides a valuable perspective to identify rare variants, as their genetic isolation can lead to an increase or a decrease in the prevalence of rare pathogenic variants[1–3]. A good example is the province of Quebec, which has experienced both founder effects and genetic drift. These phenomena contributed to an increased frequency of certain alleles within the population, but regional variations in the prevalence of those alleles can also be observed[4]. Approximately 74% of the almost 9 million present-day Quebecers are of French-Canadian origin, descending from the 8000 to 10,000 French settlers who embarked on the settlement of New France between 1608 and the British Conquest of 1759[5–7]. While the province did not adopt a policy of outright exclusion towards immigration, the subsequent waves of immigrants of diverse origins had a limited genetic impact on the established population[8]. The principal driver behind the population expansion was the pronounced natality rate. The accelerated population growth initiated a cascade of effects, including the territorial expansion of Quebec and the opening of settlements in regions previously unoccupied by European settlers[9]. The establishment of settlements in remote and geographically isolated areas further contributed to the subdivision of the population and engendered a spatially fragmented genetic structure, with distinct regional populations within the province[10,11]. This laid the foundation for the distinctive genetic makeup characterising present-day Quebecers of French descent.

An illustration of a distinct subpopulation within Quebec, considerably diverging from the major urban centers, is the Saguenay–Lac-St-Jean (SLSJ) region. SLSJ's genetics have been extensively studied because of the presence of variants that became disproportionally frequent while remaining rare in the surrounding regions[12–16]. However, some regions, presumed to be minimally affected by demographic processes, have been overlooked. One such region is Beauce, situated in Chaudière-Appalaches (Fig. 1). Beauce's settlement began in 1736 with Europeans primarily arriving from areas near Quebec City. Immigration mostly occurred in families, as for SLSJ, and the fertility in Beauce was high, with rates similar to those observed across the province, with the exception of SLSJ[17]. The Beauce region stayed open to immigration, but was difficult to access, contributing to its isolation. Settlement first started not far from Quebec City and gradually extended to further territories[18]. While this territorial expansion was primarily driven by local settlers, European immigrants also contributed. However, the proportion of the population with non-French-Canadian descent remained low and only decreased over time, dropping to less than 2% by 1931[17]. Despite those characteristics indicating a regional founder effect, the genetics of Beauce remain understudied. Current knowledge of the region's genetic landscape is limited to findings derived from simulated genomic data or clinical observations of rare disease patients[19–21]. Notably, autosomal recessive cerebellar ataxia type 1 (ARCA1), also known as ataxia Beauce type, was first described in

[1]Département des sciences fondamentales, Université du Québec à Chicoutimi, Saguenay, Québec, Canada. [2]Centre Intersectoriel en Santé Durable, Université du Québec à Chicoutimi, Saguenay, Québec, Canada. [3]Centre de recherche CERVO, Université Laval, Québec, Québec, Canada. [4]Department of Social and Preventive Medicine, Faculty of Medicine, Université Laval, Québec, Québec, Canada. [5]Department of Psychiatry and Neuroscience, Faculty of Medicine, Université Laval, Québec, Québec, Canada. [6]Projet BALSAC, Université du Québec à Chicoutimi, Saguenay, Québec, Canada. ✉e-mail: simon2_girard@uqac.ca

**Fig. 1 | Location of the regional groups mentioned in this study. a** Map of Canada and **b** zoom on southern Quebec highlighting the areas of relevance to this study (Red: Beauce, Orange: Montreal, Blue: Saguenay–Lac-St-Jean (SLSJ)). The black asterisk shows the capital of Quebec, Quebec City.

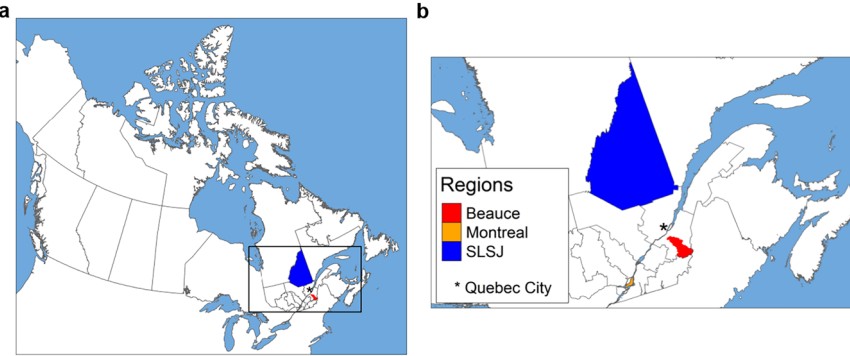

the region in 2007 and is mostly seen in the French-Canadian population[21,22].

A global portrait of the prevalence of rare variants in Beauce has never been traced, likely due to the underrepresentation of rural areas in Quebec's genetic cohorts. An overview of the rare variants that are more frequent in the region due to the settlement process could be beneficial in a clinical setting, facilitating diagnosis for patients presenting with a complex array of symptoms that could be associated with rare diseases. Therefore, this study aimed to characterise the genetics of Beauce's population and to investigate how its structure can be explained by the demographic processes using deep genealogies. We observed that Beauce exhibited heightened kinship and inbreeding coefficients and a low ancestors' diversity. Then, leveraging the genetic distinctiveness of the region, we looked at the frequency of rare pathogenic variants and identified 36 variants that are more frequent in Beauce and were likely inherited from a common ancestor and subsequently spread through drift.

## Results

### Regional genealogical structure

The evolution of the regional population's genealogical structure of Beauce, SLSJ and Montreal was compared using the mean kinship and inbreeding coefficients (Fig. 2). In Beauce and SLSJ, before 1820, the kinship increased gradually while the increase in inbreeding was steeper, reflecting the early stages of settlement where a limited number of founders had a significant contribution, exacerbated by the regional isolation. However, Montreal displayed considerably lower values during this period, likely due to its larger and more diverse set of founders. After the settling of SLSJ by individuals coming mostly from Charlevoix, both coefficients kept increasing drastically until they started to stabilise. This plateau had been observed in previous studies of the SLSJ population structure[10]. This stabilisation is similarly observed in Montreal, whereas in Beauce, the inbreeding coefficient continued to rise steadily.

Across the three regional groups, the inbreeding coefficient consistently exceeds the kinship coefficient, as the kinship coefficient is calculated between each pair of individuals, whereas the inbreeding coefficient represents the kinship between the pairs that effectively reproduced. Interestingly, the difference between the kinship and inbreeding coefficient is the lowest in SLSJ. In 1940, SLSJ inbreeding was 1.3 times higher than its kinship, while in Beauce, the inbreeding was almost 2.5 times the kinship. This accelerated increase in inbreeding in Beauce is illustrative of its effective population size (Ne), which remained low following a bottleneck approximately seven generations ago (Supplementary Fig. S1).

We then measured the ancestors' diversity ratio (ADR) at each decade for the three regional groups as an indicator of the ancestors' concentration (Fig. 3). The number of distinct ancestors gradually increased over time for each region. Montreal consistently shows the highest diversity, reflecting its position as a major urban center. In contrast, Beauce and SLSJ show considerably lower ADR throughout most of the observed period. The low ADR in Beauce is driven by the relatively frequent occurrence of a broad set of

ancestors, whereas in SLSJ, it is driven by the presence of a few 'super-ancestors' who appear at an exceptionally high number of times (Supplementary Fig. S2).

### Regional genetic structure

The present-day population structure of Quebec and its regions was assessed using principal component analysis (PCA) (Supplementary Fig. S3) and uniform manifold approximation and projection (UMAP) (Fig. 4) on the genotype data to identify clusters based on ancestry. These analyses revealed distinct regional clusters, including clusters associated with recent immigrants born outside of Canada. Cluster one, mostly composed of individuals from the SZ-BP cohort recruited in Beauce, was entirely separated from the fifth cluster, the main one representing metropolitan Quebec areas (UrbanQc). Cluster two, associated with SLSJ ancestry, was also distinguishable from the other regional populations, as expected from previous studies[1,16,19].

Density-based spatial clustering of applications with noise (DBSCAN) identified 20 distinct clusters while excluding outliers, thereby refining the population structure. We inferred the ascendance based on the recruitment region or birthplace of the individuals within the clusters. Multiple clusters were associated with metropolitan areas; however, only the largest of these clusters was retained for analysis. Those smaller clusters may represent more admixed individuals or undefined regional populations.

### Rare pathogenic founder variants

Among the whole-genome sequencing (WGS) data of 317 individuals in the Beauce cluster, we identified 43 rare pathogenic variants with a relative frequency difference of at least 10% compared to the UrbanQc cluster. For most variants, carriers were found in both cohorts (CaG and SZ-BP) within the Beauce cluster (Supplementary Fig. S4). For 36 variants, more than half of the carriers were sharing identical-by-descent (IBD) segments around the variant's position, suggesting an introduction in the region by a common ancestor. They were therefore described as founder variants in the Beauce cluster (Table 1 and Supplementary Data 1). The remaining seven likely arose from multiple introductions as assessed by the insufficient IBD sharing among the carriers (Supplementary Data 1).

Carrier rates in Beauce varied from 1/11 to 1/74 and the highest carrier rate observed in UrbanQc is 1/59 for the variant NM_001048174.2:c.1103 G > A, which also exhibits the highest minor allele frequency (MAF) among non-Finnish Europeans from gnomAD. Eleven variants were virtually absent from UrbanQc while exhibiting carrier rates from 1/21 to 1/41 in Beauce.

## Discussion

The first objective of this study was to characterise the population structure of Beauce using extensive genealogical data. Our findings provide insight into how the demographic history of Beauce has shaped its genetic profile, leading to a clear differentiation from the urban areas of Quebec. This structure is evident in the kinship and inbreeding coefficients in Beauce, which gradually increased following the initial settlement in 1736. For

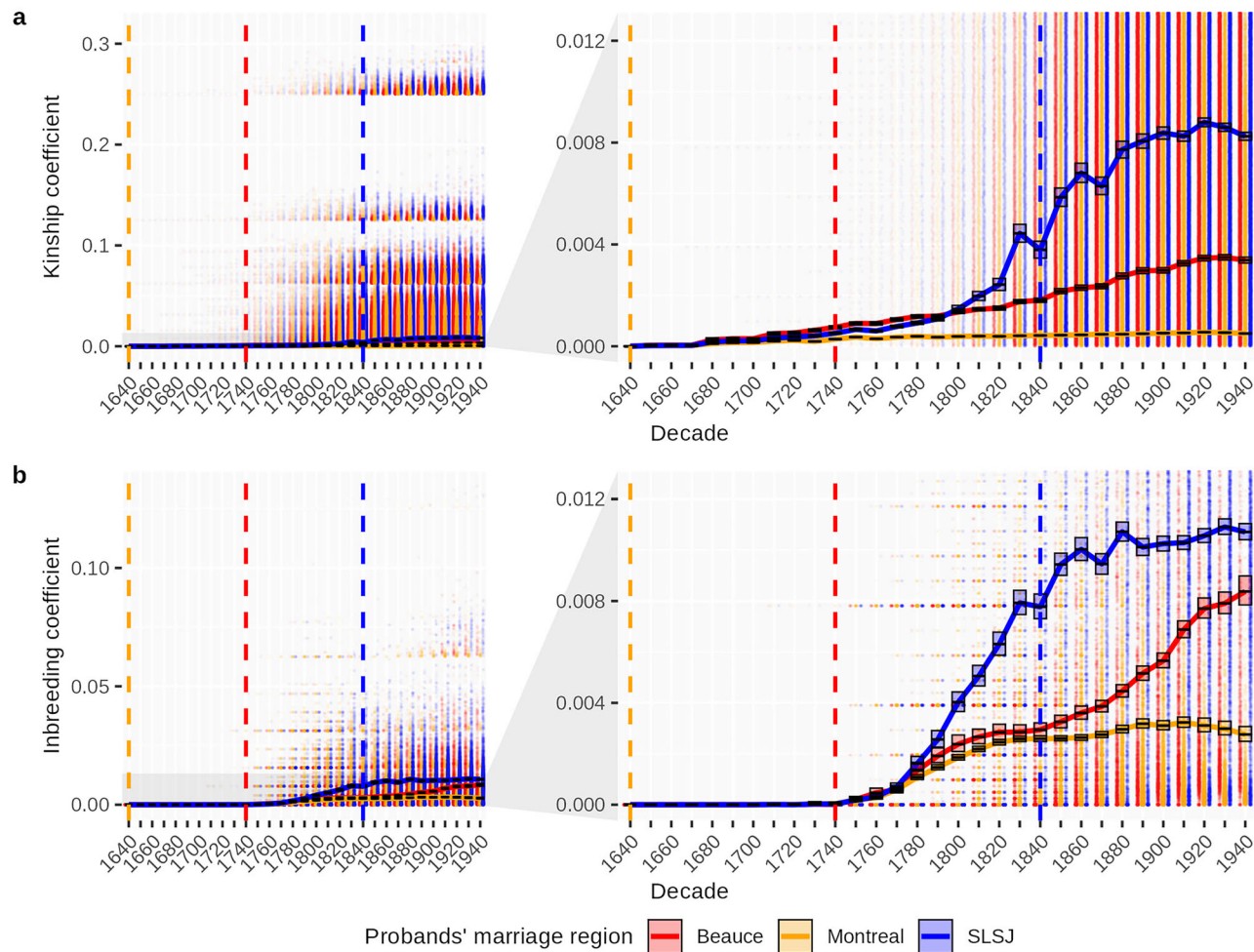

**Fig. 2 | Evolution of kinship and inbreeding coefficients across regional groups. a** Kinship coefficients and **b** inbreeding coefficients of all the ancestors of each regional group, calculated per decade. Black lines represent the mean values, boxes represent 95% confidence intervals estimated using 1000 simulations with the phiCI and fCI functions from GeneaKit and jitters correspond to individual values. Vertical dotted lines represent the decade of the beginning of the settlement for each region (Red: Beauce, Orange: Montreal, Blue: Saguenay–Lac-St-Jean (SLSJ)).

Beauce and SLSJ, settlers primarily originated from a small number of founding families coming from surrounding areas, leading to heightened kinship, but their demographic history diverged. We observe a pre-settlement structure emerging in SLSJ before 1840, while most of the ancestors were located in Charlevoix[10,23]. SLSJ then experienced a rapid demographic expansion, resulting in elevated kinship and inbreeding coefficients[10,11]. In contrast, Beauce's population growth was more gradual, with a fertility rate comparable to what was observed in the other regions of Quebec[17]. The relative geographic isolation of Beauce still led to elevated kinship and inbreeding coefficients since marriage patterns were largely confined to the community due to limited immigration[18,24]. In regions where gene flow is limited, there is a direct relationship between time and distant inbreeding[25]. Over time, such isolation can result in a reduced genetic diversity within the population[26], as seen in Beauce, where an increasing number of common ancestors among individuals led to an increase in kinship. This is also evidenced in the genetic data with a Ne which remained low after the bottleneck experienced eight generations ago during the beginning of the settlement. As individuals with increasingly closer kinship remain in the region and reproduce together, the inbreeding coefficient rises gradually[27]. Unlike SLSJ and Montreal, where inbreeding stabilised over time, Beauce saw a continued increase throughout the observed period. With a population less than half the size of SLSJ[28] and a lower Ne, individuals in Beauce were more likely to reproduce with closer relatives, contributing to the ongoing rise in inbreeding coefficients.

While inbreeding is generally higher than kinship in every regional population of Quebec, the difference between those measures is greater when closely related individuals produce a descendance[29]. The higher the kinship is between the parents of an individual, the higher their inbreeding coefficient will be[30]. After the end of the nineteenth century, there was a larger disparity between the kinship and inbreeding coefficients in Beauce and Montreal compared to SLSJ. Beauce saw a higher rate of reproduction among closer ancestors compared to SLSJ. Vézina et al.[31] already established that the close inbreeding in SLSJ is the lowest in the province, while the eastern regions like Beauce show the highest close inbreeding coefficients. In Montreal, even if the overall kinship and inbreeding are the lowest in the province, the close inbreeding is almost twice as high as in SLSJ[31], which explains why there is a greater disparity between the inbreeding and kinship coefficients than in SLSJ. Those observations are also evidenced in the evolution of the Ne inferred on the IBD segments. As expected, the urban regions present the highest Ne.

Despite differences in their demographic processes, Beauce and SLSJ share similar trends, such as low ADR throughout the observed period of the present study. The low ADR observed in those regions indicates origin from a limited number of initial settlers and limited interregional admixture, increasing the likelihood of shared ancestry among individuals. This can partly explain the high kinship coefficient of Beauce and SLSJ observed in Fig. 2. In contrast, we found a high diversity of ancestors in Montreal, where constant immigration brought individuals not only from other regions of

**Fig. 3 | Evolution of the ancestors' diversity ratio across regional groups.** Ancestors' diversity ratio of each regional group, calculated per decade. Black lines represent the mean values, boxes represent 95% confidence intervals derived from 1000 bootstrap resampling of 7445 probands and jitters correspond to individual values. Vertical dotted lines represent the decade of the beginning of the settlement for each region (Red: Beauce, Orange: Montreal, Blue: Saguenay–Lac-St-Jean (SLSJ)).

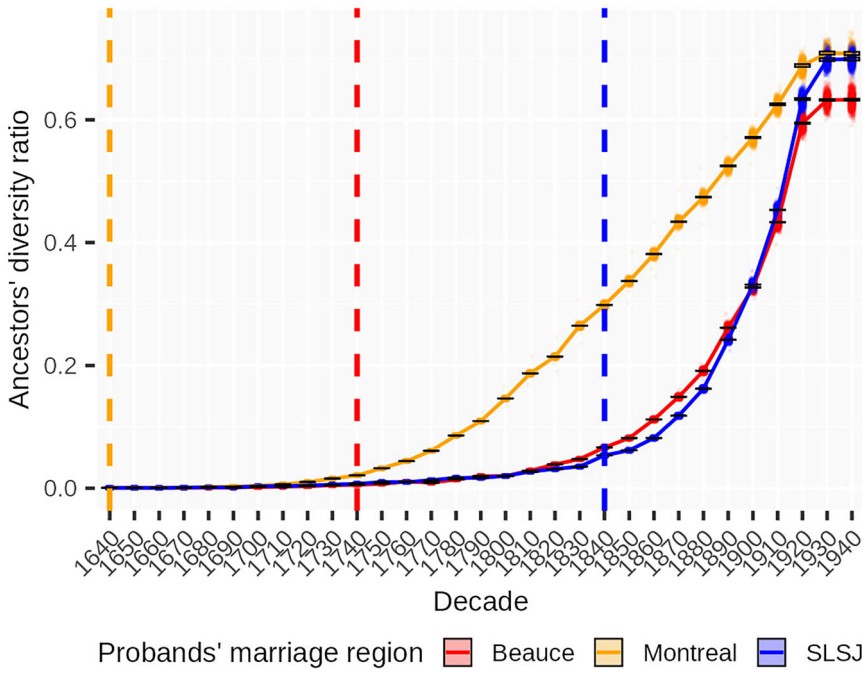

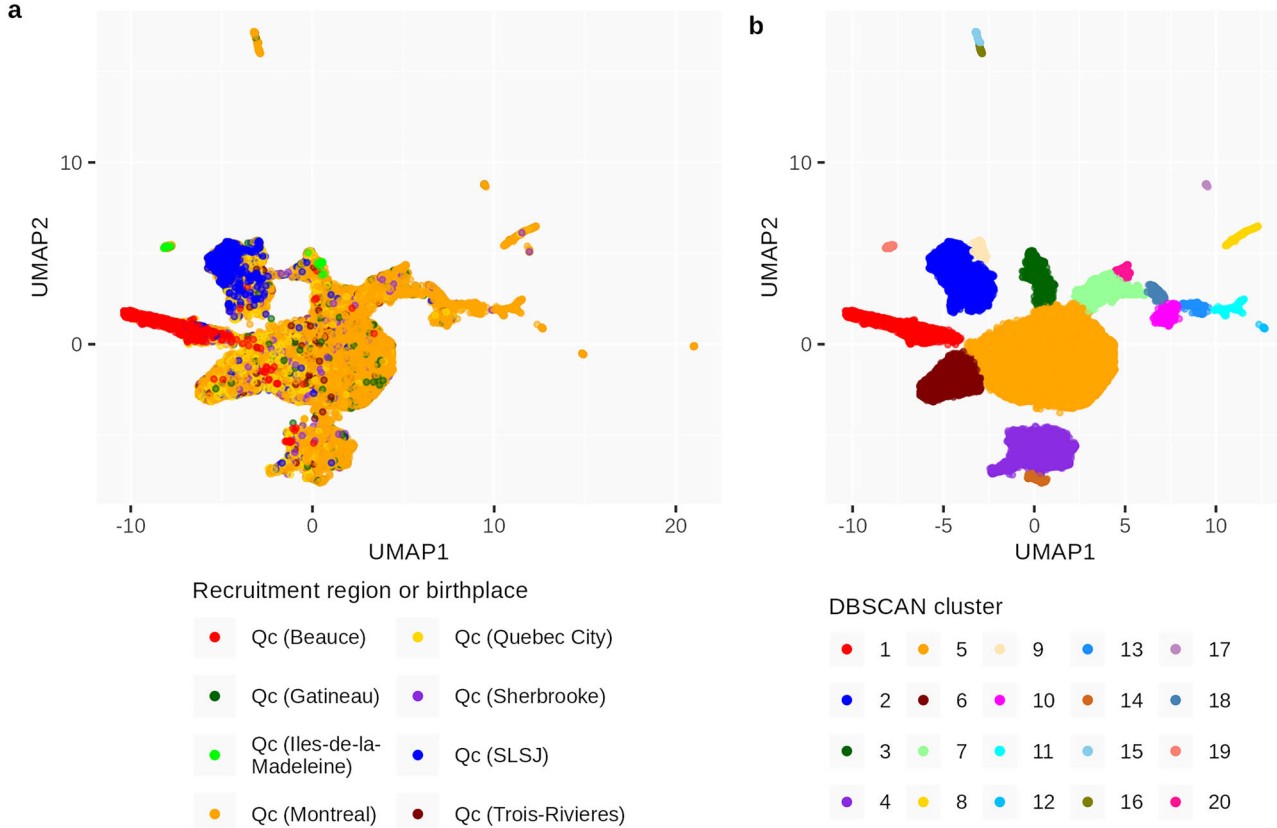

**Fig. 4 | Genetic clusters after UMAP for dimension reduction.** Two-dimensional UMAP visualisation of genotyping data based on the first 12 principal components of the PC-AiR (Supplementary Fig. S3). UMAP coloured by **a** recruitment region or birthplace and **b** DBSCAN cluster assignment (Red: Beauce cluster, Orange: UrbanQc cluster, Blue: Saguenay–Lac-St-Jean (SLSJ) cluster).

**Table 1 | Rare pathogenic founder variants with a relative frequency difference of at least 10% in Beauce compared to UrbanQc and a carrier rate of 1/25 and higher**

| HGVS name | Disease (ClinVar ID) | MAF NFE | CR Beauce | CR UrbanQc |
|---|---|---|---|---|
| NM_001199251.3:c.67 A > G | Chronic atrial and intestinal dysrhythmia (162627) | 0.0002 | 1/11 | 1/127 |
| NM_001698.3:c.656-2_656-1del | 3-methylglutaconic aciduria type 1 (1067674) | 0 | 1/12 | 1/889 |
| NM_138694.4:c.6793 C > T | Autosomal recessive polycystic kidney disease (1946278) | 0 | 1/12 | 1/221 |
| NM_025074.7:c.370 C > T | Fraser syndrome type 1 (197861) | 0.0002 | 1/13 | 1/445 |
| NM_000022.4:c.956_960del | Severe combined immunodeficiency (193544) | 0.0002 | 1/15 | 1/222 |
| NM_002225.5:c.932 C > T | Isovaleryl-CoA dehydrogenase deficiency (100060) | 0.0012 | 1/16 | 1/889 |
| NM_015665.6:c.856 C > T | AAA syndrome (1322889) | 0.0001 | 1/16 | 1/445 |
| NM_000521.4:c.1510 C > T | Sandhoff disease (3884) | 0 | 1/21 | 1/889 |
| NM_001365999.1:c.6724 C > T | Developmental and epileptic encephalopathy type 18 (429775) | 0 | 1/21 | 0 |
| NM_020458.4:c.1001+3_1001+6del | Gastrointestinal defects and immunodeficiency syndrome type 1 (50608) | 0 | 1/21 | 1/296 |
| NM_024570.4:c.529 G > A | Aicardi-Goutieres syndrome \| Cerebral palsy (1262) | 0.0024 | 1/21 | 1/74 |
| NM_025137.4:c.6598 A > T | Hereditary spastic paraplegia type 11 \| Amyotrophic lateral sclerosis type 5 \| Charcot-Marie-Tooth disease axonal type 2X (572602) | 0 | 1/21 | 1/445 |
| NM_182961.4:c.15918-12 A > G | Autosomal recessive cerebellar ataxia type 1 (2326) | 0 | 1/21 | 1/445 |
| NM_000785.4:c.262del | Vitamin D-dependent rickets type 1 A (1664) | 0 | 1/25 | 0 |
| NM_001042472.3:c.1054 C > T | PHARC syndrome (27) | 0 | 1/25 | 0 |
| NM_173477.5:c.113 G > A | Usher syndrome type 1 G (2918) | 0 | 1/25 | 0 |

*CR* carrier rate, *MAF* minor allele frequency, *NFE* Non-Finnish Europeans. All variants are autosomal recessive.

Quebec but also from other countries[32,33], leading to a higher Ne. The ancestors' diversity of the regions has been inferred in previous studies through the distribution and frequency of family names, as both are known to correlate[26,29]. Bouchard et al.[34] analysed the relative frequency of the most common family names in each region to estimate ancestral diversity. Their findings revealed that Montreal exhibited the highest ancestral diversity, while SLSJ and Beauce ranked lower, indicating reduced diversity. This was also reflected by the depth of regional ancestral roots, which represents the number of generations needed to find a majority of ancestors coming from outside of the region[24]. The low ADR we observed in Beauce is consistent with its high intraregional kinship and the gradual increase in its inbreeding coefficient, highlighting a population structure with significant shared ancestry.

The measure of ADR is largely influenced by the occurrence of ancestors, which is driven by the demographic expansion process. The population growth in SLSJ was five times greater than in the other regions of Quebec, including Beauce[17,35]. Being on the front of this rapid wave of expansion conferred a high reproductive success to the initial settlers of SLSJ, who left an important number of descendants[36]. The reduced ADR in SLSJ reflects this demographic pattern. Few ancestors have extremely high occurrences, while the rest appear at low frequencies (Supplementary Fig. S2). In contrast, the gradual localised population growth in Beauce led to a different pattern, where most ancestors exhibit consistently high frequencies, resulting in a similarly reduced ADR. This also resulted in an elevated kinship in Beauce, as individuals share a high number of common ancestors due to the low Ne. While elevated, the kinship and inbreeding coefficients in Beauce are still lower than what is seen in SLSJ because of the more gradual population expansion.

As previously demonstrated, the genealogical structure of a population mirrors its genetic structure[10,11,19]. Thus, the high kinship and inbreeding coefficients and the low ADR in Beauce are indicative of a population where individuals share high proportions of their genomes. Rare pathogenic variants could have been introduced by initial founders, then spread through drift in the population, leading to a higher frequency than what can be observed in populations that did not go through this regional founder effect.

The second aim of this study was to explore the frequency of rare pathogenic variants in Beauce compared to more admixed regions to identify variants present at higher frequencies. More precisely, we wanted to identify variants with a higher frequency in Beauce than in the metropolitan areas of Quebec because of the regional founder effect.

We first defined population groups using a UMAP visualisation to identify compact clusters of individuals with similar genetic backgrounds[37,38]. The distinct regional clusters demonstrate that the Quebec population cannot be treated as a single genetic entity, as it comprises multiple, regionally defined subpopulations with unique genetic and demographic histories. This clustering enhanced the identification of rare pathogenic variants concentrated in specific regional groups, where their increased frequency facilitated detection[1,16,39]. Using this method, we identified 43 rare pathogenic variants with a relative frequency difference of at least 10% between the Beauce and UrbanQc clusters. In addition to accounting for their high frequency, we examined the IBD segments sharing between carriers and identified 36 rare pathogenic variants for which the elevated frequency and IBD sharing in Beauce are consistent with the effects of a regional founder event and demographic isolation.

Among those, 20 were previously reported in Quebec. Notably, NM_182961.4:c.15918-12 A > G was identified as founder in Beauce, exhibiting a carrier rate of 1/21. In Eastern Quebec, it is the most frequent of the seven ARCA1 variants reported among individuals of French-Canadian descent, with a minimum carrier frequency reported at 1/134[21,40]. Since previous studies focused on a broader region, they obtained a much lower carrier rate than what we observed. Including metropolitan areas in their study likely diluted the frequency of the variant because, as shown in the present study, this variant is around 21 times more frequent in Beauce than in urban populations of Quebec. This elevated frequency was previously attributed to a founder effect in the French-Canadian population based on the prevalence of the disease and genealogical reconstruction[21] and our analysis of IBD sharing reinforced its founder origin in Beauce. No other known variants for ARCA1 were found in the Beauce cluster, but three were present at low MAF in UrbanQc.

For one of our previously reported founder variants, the IBD sharing between the carriers in Beauce and UrbanQc indicates a shared founder event. NM_001199251.3:c.67 A > G had already been suspected to be more frequent among French-Canadians due to a founder effect[41]. We observe it at a much higher frequency in Beauce, with a carrier rate of 1/11. The other previously described founder variants were only founder in Beauce based on

**Table 2 | Description of the genealogical data for each regional group**

| Region | Number of probands | Number of ancestors |
|---|---|---|
| Beauce | 7445 | 61,569 |
| Montreal | 10,048 | 239,011 |
| SLSJ | 9655 | 78,714 |

the insufficient IBD sharing among the UrbanQc carriers. For NM_020458.4:c.1001+3_1001+6del, a variant associated with gastrointestinal defects and immunodeficiency syndrome type 1, a founder effect was suspected among French-Canadians because of its high frequency[42], and it was described as founder in SLSJ, with a carrier rate of 1/56[16]. We observed a higher carrier rate in Beauce with 1/21. We also described a variant associated with Tay-Sachs disease, NM_000520.6:c.1274_1277dup, to be more frequent in Beauce because of the founder effect. Multiple Tay-Sachs variants are reported in Quebec, with regional differences in frequency. While this variant was previously described as more frequent among French-Canadians, it was never traced back to a specific region[43].

A recent study in SLSJ identified 80 rare founder pathogenic variants and approximately half were previously documented[16]. Unlike SLSJ, Beauce does not benefit from extensive genetic data and a well-documented prevalence of rare diseases. By applying a similar methodological approach to Beauce, we provide reliable findings despite the limited literature on the prevalence of genetic diseases in this region. We identified 16 variants that appear to be founder in Beauce and were not previously documented in Quebec, illustrating the pertinence of population genetics studies to document the increased risk of rare diseases for certain populations. Two of the most frequent founder variants in Beauce, NM_000022.4:c.956_960del, associated with severe combined immunodeficiency, and NM_001698.3:c.656-2_656-1del, associated with 3-methylglutaconic aciduria type 1, have not been previously documented among French-Canadians despite high regional carrier rates. Additionally, four unreported founder variants were virtually absent from UrbanQc with carrier rates of 1/29 to 1/49 in Beauce. Interestingly, NM_001048174.2:c.1103 G > A, a relatively frequent variant associated with MUTYH-related cancer predispositions among individuals of European descent[44], was identified as founder in Beauce, but not UrbanQc because of the insufficient IBD sharing, despite its spread through founder events that occurred during early European history[45]. It was, to our knowledge, never reported among individuals of French-Canadian descent.

While most founder variants are recessive, we identified three dominant founder variants, only one of which was previously reported. NM_007194.4:c.1100del is a frequently occurring variant in individuals of French-Canadian descent with a history of breast cancer[46]. The two others were seen to act as either dominant or recessive variants, which could explain why they remained unnoticed. NM_001039213.4:c.1186 A > G, associated with rare genetic deafness, is globally rare and was found in 1/29 individuals in the Beauce cluster. It might have remained unnoticed in the region due to limited genetic testing for rare genetic deafness and to the milder phenotype associated with the dominant form[47,48]. Similarly, NM_000228.3:c.2842del was never reported in the province despite having a carrier rate of 1/49 in Beauce. In its recessive form, it is associated with epidermolysis bullosa junctional (type 1 A or 1B), but its dominant form is associated with amelogenesis imperfecta (OMIM #150310).

This study is limited by the sample size of the Beauce WGS cohort, which may not be large enough to accurately represent carrier rates across the entire general population of the region. As a result, our findings should be interpreted as a preliminary exploration rather than a definitive characterisation. However, since individuals were not selected based on a rare disease diagnosis, and since the clustering was based on the genetic similarities between individuals, it could still enhance regional pathogenic variant discovery. We also acknowledge that the relative frequency threshold of at least 10% is somewhat conservative and may have limited the number of

variants detected and possibly excluded variants of potential interest. While the results provide valuable insights into the rare variants with a higher frequency in the region, further studies with larger, more representative cohorts will be necessary to better estimate global carrier rates. Although, since the present-day Beauce population results from a regional founder effect, we might need a less important sample size to be representative of the population than for more admixed regions.

In conclusion, this study provides a genealogical and genetic characterisation of Beauce, a regional population shaped by a limited number of founding families and geographic isolation. Those factors led to elevated kinship and inbreeding coefficients, a low ADR, and a unique genetic structure, as evidenced by a distinct cluster of individuals with Beauce ancestry in the UMAP visualisation of genotype data. Within this cluster, we identified 36 rare pathogenic variants with a higher frequency in Beauce and an IBD sharing among carriers that is indicative of an inheritance from a common ancestor. Of those, 20 variants were previously described in the province. This study highlights the demographic and genetic distinctiveness of Beauce and underscores the pertinence of fine-scale genetic analyses to uncover region-specific rare variants, even within a population that was previously thought to be homogenised by the initial founder effect. Our approach offers a powerful framework for the identification of rare variants and has the potential to facilitate population health management. By focusing on smaller cohorts of individuals who share a common ancestry, this method enhances the discovery of rare variants as it increases their frequency. Moreover, applying this approach to existing large cohorts could uncover other populations with distinct genetic structures that may have been understudied, thereby enhancing the value of existing cohorts while remaining cost-effective[16,39,49].

## Methods
This study was approved by the *Université du Québec à Chicoutimi* (2024-1396) and the *Centre Intégré Universitaire de Santé et de Service Sociaux de la Capitale-Nationale* (2019-1597, NSM) ethics boards. Written informed consent was obtained from all adult participants or the parents for participants under 18 years of age. All ethical regulations relevant to human research participants were followed.

### Genealogical data and cleaning
The genealogical data used in this study were obtained from the BALSAC database[50]. BALSAC is a repository containing Catholic marriage records and civil registry data used to reconstruct genealogies. The available data range from the beginning of the European settlement in the early 1600s to 1965.

To describe the regional structure of the populations, we selected all individuals married in Beauce, SLSJ and Montreal between 1935 and 1960 and reconstructed their genealogies going back over three centuries, thus covering almost the entirety of Quebec's post-European settlement history. We then performed a quality control based on the completeness of the genealogies, which corresponds to the proportion of known ancestors on all expected ancestors for every generation[30]. For Beauce and SLSJ, only genealogies with perfect completeness up to the third generation were included. For Montreal, since the overall completeness considerably dropped after the fourth generation, only the genealogies complete to the fifth generation were retained. All three regions then showed similar completeness throughout generations (Supplementary Fig. S5). Then, we removed first-degree relationships among the probands to avoid redundancies and, to ensure that the genealogies accurately reflect the regional ancestry, probands who immigrated recently (within three generations) were removed. The final dataset is described in Table 2.

### Genealogical analyses
All genealogical analyses were conducted with the Python library GeneaKit version 0.1.0. For each regional group, we measured the evolution of the averaged kinship and inbreeding coefficients, as well as the ADR for each decade from 1640 to 1950. Kinship being directly influenced by the presence

**Table 3 | Description of the genomic data available**

| Cohort | Recruitment region | Data type | Sample size |
|---|---|---|---|
| CARTaGENE | Gatineau, Montreal, Quebec City, SLSJ, Sherbrooke and Trois-Rivieres | Genotyping | 28,248 |
| | | Whole genome sequencing | 2184 |
| Eastern Quebec Schizophrenia and Bipolar Disorder Kindreds Study | Beauce, SLSJ and Iles-de-la-Madeleine | Genotyping | 1200 |
| | | Whole Genome Sequencing | 500 |

of close relatedness between pairs, relationships of the first degree were removed for each decade. We calculated 95% confidence intervals over 1000 simulations.

We assessed the ADR to represent the concentration of the ancestors in each regional group. To measure the ADR, we computed two metrics: the number of ancestors present in the genealogy of all probands, and their occurrence. The occurrence represents the overall count of all ancestors identified, including multiple apparitions, while the number of ancestors reflects the distinct individuals counted without repetition[31]. We then computed the ratio of the number of ancestors to the occurrence of ancestors as follows:

$$ADR = \frac{Number\ of\ distinct\ ancestors}{Ancestors'\ occurrence}$$

To ensure comparable measurements between the regional groups, we used a bootstrap approach where we randomly selected a subset of 7445 probands from each regional group. The replicated sample size corresponds to the number of probands in the smallest regional group. This process was repeated 1000 times, and we estimated the mean ADR and 95% confidence intervals.

**Genotyping data and cleaning**

The genetic data in this study include the CARTaGENE cohort (CaG) and the Eastern Quebec Schizophrenia and Bipolar Disorder Kindreds Study cohort (SZ-BP). The CaG cohort comprises genotyping data for 29,337 individuals aged between 40 and 69 who were recruited based on their residing regions[51]. Recruitment occurred between 2009 and 2015 and was performed regardless of the individuals' birthplaces. Participants residing in six metropolitan areas (Montreal, Quebec City, Trois-Rivières, Sherbrooke, Gatineau, Saguenay) were randomly chosen to be broadly representative of the population based on provincial health insurance registries.

Individuals were genotyped in different waves on different genotyping arrays (Omni 2.5, GSAv1 + Multi disease panel, GSAv1, GSAv2 + Multi disease panel, GSAv3 + Multi disease panel, GSAv2 + Multi disease panel + addon and Affymetrix Axiom 2.0). A more complete description of the genotyping technologies used and a breakdown of the number of individuals per array are available in the CaG documentation (https://cartagene.qc.ca/files/documents/other/Info_GeneticData3juillet2023.pdf). We chose to exclude the Affymetrix chip due to its poor SNP intersection with the others.

To include individuals from the Beauce region, we added participants from the SZ-BP cohort. The SZ-BP cohort comprises 1120 individuals distributed in 48 multigenerational families affected by schizophrenia and bipolar disorders, as well as 80 additional unrelated patients[52,53]. All individuals were recruited in Beauce, SLSJ and Iles-de-la-Madeleine. Genotyping data is available for all individuals and was carried out in two waves using DNA extracted from immortalised lymphocytes or fresh blood by affinity column (Midi prep Qiagen). 585 subjects were genotyped at 651,692 autosomal SNPs with the Illumina Infinium Human OmniExpress array and 615 subjects were genotyped at 691,719 SNPs with the Illumina Global Screening Array. More details are provided in Bahda et al.[54]. Genotyping

data was lifted over from the GRCh37 to the GRCh38 build using the LiftOver Genome Browser software.

Each dataset was cleaned independently using plink/1.9b_5.2-x86_64. SNPs missing in at least 5% of the subjects were excluded, and individuals with a genotype missingness of at least 5% were removed. SNPs that did not conform to the Hardy-Weinberg Equilibrium with a $P$-value threshold of $10^{-6}$ were also removed. All genotyping data was merged based on the intersection of positions. After the merge, the same quality control criteria were reapplied. SNPs were then pruned using a window size of 50, a step of 5 and a minimum correlation threshold of 0.2 and alleles with a MAF below 5% were excluded. The clean dataset comprises 29,448 individuals with 60,962 SNPs.

The content of both samples is summarised in Table 3.

**Ancestry-based clustering**

Since individuals in the CaG cohort were recruited according to their residing region instead of their birthplace, we formed population clusters based on their genetic ancestry using the genotyping data. We first excluded individuals born outside of Canada, leaving us with 26,649 genotyped individuals. To obtain principal components (PC) that capture ancestry and not close relatedness, we performed a PCA on the genotyping data using the PC-AiR function from the R library GENESIS/2.28.0. Individuals related up to the third degree were identified based on the proportion of IBD sharing. IBD segments were inferred on the phased genotypes with refinedIBD version 17Jan20[55] within Beagle version 18May20[56]. The IBD segments were detected using a sliding window of 40 SNPs, a log of odds threshold of 3, and a minimal length of 2 cM with a sensitivity scale of 10 to retain segments in familial data. Segments were then merged using the merge-ibd-segments 17Jan20.102 tool[55]. We also used the genealogies to ensure that no related individuals were missed. We identified groups of individuals sharing a proportion of at least 0.125 IBD segments or with a genealogical kinship of at least 0.0625 and randomly kept only one individual per group. This left us with a set of 1068 related individuals and 25,581 unrelated individuals, which we used to get our ancestry-representative PCs. Using the first 12 PCs, a UMAP was performed with the R umap/0.2.10.0 library[57]. We selected the first 12 PCs based on the third cutoff point identified in the scree plot using the R library changepoint/2.2.4 and because it allowed a visual separation of individuals recruited in Beauce (Supplementary Figs. S3 and S6). The UMAP was performed to emphasise the local genetic structure within the data and was done on the first 12 PCs instead of the genotyped data to remain computationally efficient[37].

DBSCAN was performed on the UMAP using the dbscan library[58] to identify our ancestry-based clusters[38,59]. The size of the epsilon neighbourhood was set to 0.30, and the minimum number of points in the neighbourhood was set to 50. Those parameters were chosen after several iterations with different combinations to ensure the identification of meaningful clusters while minimising the inclusion of noise.

This resulted in 20 distinct clusters, and their origin was assigned based on the recruitment region of most individuals within the clusters. Clusters one and five were kept for further analysis. They respectively corresponded to Beauce and the urban regions of Quebec (UrbanQc). This left us with 13,360 genotyped individuals. The Beauce cluster comprised 1362 individuals, of whom 44% were recruited in Beauce (which corresponds to 93% of all individuals recruited in Beauce). The rest were recruited mostly in metropolitan areas without information about their birthplace. The UrbanQc cluster comprised 11,998 individuals recruited mostly in urban centers (Supplementary Fig. S7). A cluster of individuals from SLSJ was also identified ($n = 3162$). We inferred effective population size (Ne) for the Beauce, SLSJ and UrbanQc clusters using the ibdne software (version 23Apr20.ae9) with default parameters. IBD segments were inferred on the phased genotypes using hap-ibd (version 1.0, 15Jun23.92 f).

**Whole-genome sequencing data and cleaning**

To identify rare variants with higher frequencies in the Beauce cluster, we used the whole-genome sequencing (WGS) data available for some

individuals in the SZ-BP and CaG cohorts (Table 3). WGS data was available for 2184 genotyped CaG individuals. Unrelated individuals were selected in 2022 if they were alive and reachable by email and had four grandparents originating from the same country (Canada ($n = 1889$), Morocco ($n = 132$), Haiti ($n = 163$)). Sequencing was performed at the *Centre d'Expertise et de Service de Génome Québec* using Illumina NovaSeq S4 platform with a PCR-free, paired-end $2 \times 150$ base pairs protocol. The data analysis was carried out with the Illumina DRAGEN platform. Variant calling was performed jointly across genomes, generating a multi-sample VCF per chromosome. A full description of the WGS data and quality control is available in the CaG documentation (https://cartagene.qc.ca/files/documents/other/Info_GeneticData3juillet2023.pdf).

WGS data was also available for 500 genotyped individuals from the SZ-BP cohort. DNA was extracted from stored frozen blood leucocytes and was prepared using the TruSeq DNA PCR-free library. The sequencing was performed with the short-read Illumina NovaSeq S4 flowcell version 1.5. We used the DRAGEN Germline Pipeline v3.10.4 on the Illumina BaseSpace to align the FASTQ on the hg38 Human reference genome and to call variants. We then performed the same quality control that was conducted on the CaG WGS data.

The gVCF files from the SZ-BP and CaG Canadian cohorts (to match the ancestry of the SZ-BP subjects) were joint called using the Population explorer WebApp (https://popex.dragen.illumina.com/) to produce a joint VCF file. This produced a single combined VCF file of 79,000,100 positions (chr 1 to X) for 2389 subjects. WGS are available for 317 genotyped individuals in the Beauce cluster and 893 genotyped individuals in the UrbanQc cluster.

### Identification of rare pathogenic variants with higher frequency in Beauce
We selected variants in the ClinVar database[60] (version of June 24th, 2024) following the same criteria as Michel et al.[16]. Only variants classified as pathogenic, likely pathogenic, or conflicting, as well as SNPs, insertions and deletions were included. Variants with the following review status were removed: no assertion criteria provided, no classification provided and no classification for the individual variant. Additionally, all variants referenced as founder variants in previous studies on the Quebec population[12–15] were included, regardless of their status on ClinVar, leaving us with a total of 240,716 variants. We determined the MAF of the selected variants in the Beauce and UrbanQc clusters using Plink. Since most of the individuals in the Beauce cluster are from the SZ-BP cohort ($n_{CaG} = 69$, $n_{SZ-BP} = 248$), we compared the MAF of all the rare pathogenic variants between the cohorts to ensure the absence of a batch effect (Supplementary Fig. S8). To identify the variants at higher frequencies in Beauce, we calculated the relative frequency difference, which corresponds to the relative difference between the MAF. The relative frequency difference is calculated as follows:

$$Relative\ frequency\ difference = \frac{MAF_{Beauce} - MAF_{UrbanQc}}{MAF_{Beauce}}$$

Only variants present in at least five carriers and reaching a relative frequency difference of at least 10% in the Beauce cluster compared to the UrbanQc cluster were considered. We set the five carriers' threshold to ensure that only variants with sufficient representation across families were included in the analysis. A liberal threshold of 10% was chosen as suggested by Michel et al.[16]. Given the limited literature related to the regional variants with a higher frequency in Beauce, we preferred using the same cutoff point despite having a smaller sample size to avoid the risk of having false positives. MAF were also obtained for non-Finnish Europeans from gnomAD v4.1.0[61]. Because our study focuses on rare variants, two variants with a MAF exceeding 5% were excluded. Finally, carrier rates were calculated using the proportion of heterozygous individuals for each variant in the Beauce and UrbanQc clusters. Heterozygous individuals were identified using plink. To ensure that MAF and carrier rates were not confounded by the relatedness within the SZ-BP cohort, we calculated them on a maximally unrelated

sample. Values calculated on the whole sample are available in Supplementary Data 2.

### Classification of rare pathogenic variants with higher frequency in Beauce
We used IBD segments shared between the carriers of each variant to identify the variants that are more frequent in the Beauce cluster due to a regional founder effect. For each variant with a higher relative frequency in Beauce, we examined the IBD sharing between the carriers and determined the proportion of pairs sharing at each genomic position. We considered the variant as founder if at least 50% of the pairs of carriers shared IBD around the variant's position. To ensure that the variants were indeed linked to a founder effect and not biased by the related individuals present within the sample, we inferred relationships between the carriers using the total of IBD sharing and the genealogies when available. Variants of familial origin, defined as variants carried by less than five unrelated individuals (proportion of total IBD sharing <0.125 or genealogical kinship <0.0625), were excluded. Variants that did not fit either category were classified as variants with multiple introductions in the population.

### Statistics and reproducibility
To ensure reproducibility, we provide descriptions of the analytical procedures, including data inclusion and exclusion criteria, software version and parameters. All software used are publicly available and listed with specific version numbers, and all custom code is available on GitHub.

Genealogies were reconstructed using Catholic marriage records and civil registry data from the BALSAC database. We included all individuals married in Beauce, SLSJ and Montreal between 1935 and 1960. All genealogical analyses were performed using the Python library GeneaKit version 0.1.0, including the filtering of individuals with insufficient completeness or related individuals. The sample size for the genealogical analyses represents the maximum number of individuals passing quality control for each regional group ($n_{Beauce} = 7445$; $n_{Montreal} = 10,048$; $n_{SLSJ} = 9655$). For genetic analyses, we used data from existing cohorts. Individuals were excluded if they failed the standard quality control described earlier, if they reported being born outside of Canada, or, later in the analyses, if they were not included in the clusters of relevance to our study. Dimensionality reduction was performed using the PC-AiR function (GENESIS R package, v2.28.0) and the umap function from the umap package (v0.2.10.0) and clustering with dbscan (v1.2.2). The genotyping data was retained for three clusters ($n_{Beauce} = 1362$; $n_{Montreal} = 11,998$; $n_{SLSJ} = 3162$) and the WGS data for two clusters ($n_{Beauce} = 317$; $n_{UrbanQc} = 893$).

### Reporting summary
Further information on research design is available in the Nature Portfolio Reporting Summary linked to this article.

### Data availability
Access to the genealogical data used in this study may be granted to researchers for scientific purposes, provided that the request aligns with the objectives of the data file. Requests must be submitted to the BALSAC Project's Researcher Services, which is responsible for managing and evaluating access applications. To request access, researchers must complete the access request form available on the BALSAC website and send it to balsac@uqac.ca. Further information is available at: https://balsac.uqac.ca/en/acces-donnees/. The aggregated data presented in Figs. 2 and 3 is available on GitHub (https://github.com/Genopop/Rare_founder_variants_Beauce).

Due to recent data protection regulations in Quebec, the sharing of raw genomic data is subject to strict constraints. As a result, data from the Eastern Quebec schizophrenia and bipolar disorder kindred study is not publicly available at this time. Efforts are currently underway to deposit the data with the *Centre québécois de données génomiques*. In the meantime, data may be made available upon reasonable request from MM. The data from the CARTaGENE cohort is publicly available via an independent data access committee by the CARTaGENE cohort (https://cartagene.qc.ca/en/

researchers/access-request.html). The source data behind Fig. 4 is also available on GitHub.

## Code availability

The code for the genealogical analyses and the founder variants identification is available on GitHub (https://github.com/Genopop/Rare_founder_variants_Beauce).

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

## Acknowledgements

The Eastern Quebec schizophrenia and bipolar disorder kindred study was funded by a Canada Research Chair (#950-200810) in psychiatric genetics, of which MM is the Chair and by Canadian Institutes of Health Research grants (#MOP-74430, #MOP-114988, #PCG-155471, and #PJT-175122). Funding for SLG was provided by the Canada Research Chair (#CRC-2022-00444) in Genetics and Genealogy. MG received scholarships from the Fonds de recherche du Québec – Santé (https://doi.org/10.69777/331979) and the Canadian Institutes of Health Research.

## Author contributions

M.G., C.M. and S.L.G. designed this study. M.G. analysed the data and wrote this manuscript. A.B. and M.M. collected the samples. JR provided biostatistics expertise. M.-C.B. provided technical assistance in the sample preparation. All authors contributed to the article and approved the submitted version.

## Competing interests

The authors declare no competing interests.
