## [Transparent Peer Review file · Communications Biology]

Rare variants and founder effect in the Beauce region of Quebec

Corresponding Author: Professor Simon Girard

Version 0:

Reviewer comments:

Reviewer #1

(Remarks to the Author)

The authors present an interesting example of identifying potential founder rare pathogenic variants in a region by combining genealogical information and population structure. For population structure, the authors used UMAP. For rare variants, the authors used information from the ClinVar database. Overall, I think the novel findings of this study should be better highlighted and the narrative could be improved. For example, the rationale for selecting rare variants is not clear and the correlation between the variants identified and their genetic background is not statistically clear. The classification of rare pathogenic variants as more frequent in Beauce does not necessarily suggest a causal relationship between regional population structure and regional rare pathogenic variants. The population structure could have been investigated in a more sophisticated way (e.g. by identifying ancestry sources, quantifying admixture proportions, etc.) in order to explain the heterogenous population structure in UMAP with further analyses. Nevertheless, this study is a nice attempt to correlate pathogenic variants with population structure (and genealogy).

Reviewer #2

(Remarks to the Author)

Review:

I had the pleasure of reading the article "Rare variants and founder effect in an understudied Quebec population" by Gagnon et al. The authors present a compelling motivation to study the founder genealogies of Beauce and Saguenay–Lac-St-Jean (SLSJ) populations (Quebec, Canada) to understand the prevalence of pathogenic rare variants. Overall, I found the article to be well written and includes a wealth of cited sources making the article flow logically and allow the authors to test their hypotheses in a thoughtful approach.

The authors use array genotyping, whole-genome sequencing, and genealogy datasets to assess specific Quebec communities for the prevalence of pathogenic variants as well as historic estimates of ancestral diversity. I think that the authors approach is sound and clever to use such complimentary datasets and techniques. I present some issues that the authors could mitigate below and some suggested analyses that could improve the scope of the paper, although not critical for publication. I think there needs to be some additional clarity over the implementation of their pathogenic variants and the specific hypotheses this line of questions entail. Overall, the authors' findings are useful to the broader scientific community, and I think that the study provides foundational clinical evidence for their study populations in the future. I think generally, founder effects are largely forgotten to affect human populations, and it was nice to see this article address the medical consequences of small founding effective population sizes.

Comments:

Line 108-109: Why not calculate 95% confidence intervals or quantiles?

Lines 113-115: I think the inclusion of familial cases of schizophrenia and bipolar disorder will skew the frequency of potentially pathogenic variants. This dataset could be used as an advantageous position to assess this conclusion. Do you prioritize these variants in any way downstream? It is not clear to me.

Line 122-125: I think it is important to include more details regarding the bioinformatics. You have multiple Illumina technologies and the variant calling/downstream analyses could be affected by this fact. Please expand the details and make sure it is the same human genome used for mapping, if not you will have to LiftOver positions.

Line 142: There is a package in R to determine the “elbow” of a scree plot that maximizes the information gained from the inclusion of multiple PCs. This might be interesting to test for your data: <https://github.com/heatherjzhou/PCAFForQTL>, <https://doi.org/10.1186/s13059-022-02761-4>

Line 149-152: This kind of seems arbitrary to me.

Line 172-173: Briefly describe how you are prioritizing ClinVar variant screening. I think this is a sticking point for me throughout the article.

Line 184: Why non-Finnish? My gut instinct would be to not exclude any populations that make up the European dataset from gnomAD. Some justification would be helpful.

Line 191-193: I wanted to see the inclusion of Ne estimates for your populations of interest. I know these methods are really for long-term evolutionary processes, but I think they can estimate accurately up until 1600-1700s. I saw this tool that uses IBD to estimate Ne and outputs confidence intervals, your work mentions merge-ibd-gaps as a processing step and this software has it built in: <https://faculty.washington.edu/browning/ibdne.html>; <https://doi.org/10.1371/journal.pgen.1007385>

Line 193: “enriched” will need to be defined, are these SNPs within windows that have an odds ratio > 3?

Figure 2: This figure is a bit blurry on my end. I think it would be helpful to show how many variants / samples were included in each step. It is a bit verbose currently though.

Results 3.1: I think it would be super interesting to include some estimates of the effective population size (Ne) of the populations in Beauce and SLSJ in a proper way. I have used the MSMC2 tool in the past that can work with VCFs in your dataset following some additional pruning but it seems highly relevant for your purposes, this could also allow you to make a statement regarding the change in historic population size relative to Montreal.

Figure 3: I was confused why there are values for Beauce and SLSJ prior to the decade beginning settlement. For SLSJ, there is a stark increase in relatedness prior to settlement and it staggers off post-settlement. While Beauce shows an increase post-settlement, implying more inbreeding in Beauce than SLSJ and generally lower Ne.

Line 261 : Some of your carrier rates are below 1/20, which would equate to a frequency of less than 5%. I am confused how this works given that there are 223 individuals in your WGS dataset (for Beauce). Also, line 180 says that “Only variants present in at least five carriers and reaching a relative frequency difference of at least 10% in the Beauce cluster compared to the UrbanQc cluster were considered”. So, I see a disconnect on how carriers were identified and I think it would help readers to make it more specific and add a section in the methods for this explicitly.

Does the 1/14 carrier rate mean that there is only 1 individual that is heterozygous for that variant? What does the denominator and numerator represent?

SuppTable 1 & Table 3 & SuppData1: Please give chromosome + position on HG38. Can you supply the relative frequency difference? I think it would be helpful to know the relative frequency difference for each variant as an additional column. For table 3, there is no variation in inheritance so you can remove that column and state it in the legend.

SuppFig 7: Can you remind briefly what is the difference between founder and multiple introductions in the legend? Also, it is a bit hard to see the shapes at the bottom, you could think about using facet_wrap(~Variant classification) to split the plot (I am assuming you are using ggplot2 in R) into the two categories instead.

General comments on bioinformatics: I admire that you were rigorous with showing which versions were used for all software.

Data availability: You need to include a GitHub or similar coding repository for all of the code and scripts for your project.

Reviewer #3

(Remarks to the Author)

"Rare variants and founder effect in an understudied Quebec population"

Overview:

Gagnon and colleagues present a population-level genomic analysis of different regions in the Quebec province. Using genealogical data, the authors show clear evidence of founder effects in Beauce and Saguenay-Lac-St-Jean, distinct from the more ancestrally diverse area of Montreal. The authors describe the history and dynamics of settlement in the area and how that aligns with the observed genealogical data. Next, the authors use an identity-by-descent approach to identify clinically relevant variants in Beauce likely arising from founders. This work is in general well conceptualised and represents the first exploratory analysis of the clinical genomic landscape of the Beauce region. However, despite the intriguing results, I have some issues with the methodology which may impact the conclusions of this manuscript.

Major comments:

1 – In Section 2.4, the authors are rightly concerned about how PC shrinkage following a projection may cloud the fine population structure. However, their decision not to adjust for the relatedness in the cohort will have impacts downstream that cannot be alleviated. For example, the majority of the Beauce cluster from the UMAP analysis is derived from the SZ-BD family cohort. The authors cannot know if this distinct clustering is a result of population structure in Beauce or due the high degree of relatedness in the selected individuals (likely both). To avoid this issue, the authors could either take a maximally unrelated subset of the SZ-BD cohort or adjust for the relatedness in the PCA via projection. Indeed, the publication the authors cite in regard to this discusses strategies to reduce PC shrinkage following projection (Privé et al., *Bioinformatics*, 2020). Regardless, the relatedness as described should not be left unaccounted for.

2 – I am not convinced that it is appropriate to treat the clinically-derived SZ-BD cohort as a population-representative group. In general, individuals with schizophrenia or bipolar disorder likely represent less than 3-4% of the general population and are enriched for rare, damaging variants. As such, it may not be surprising to see rare pathogenic variants in these individuals or their close relatives. If this cohort is to be retained, the authors should detail their attempts to mitigate the selection bias compared to the more broadly recruited CaG cohort. Otherwise, the results described may not generalise to the entire Beauce region.

3 – In Sections 2.5 and 3.3, MAFs and CRs for rare variants of interest are described to prioritise those found more frequently in Beauce. The authors should confirm that these values were calculated on an unrelated subset of the respective groups, since the high degree of relatedness from SZ-BD will incorrectly inflate the MAFs. Perhaps I am misinterpreting, but some of the pathogenic variants listed in Supplementary Table 1 are quite common in the Beauce cohort (equivalent MAF of 2-7%) but are extremely rare or absent in the well-powered gnomAD database. It doesn't seem plausible to me that, for example, ~3.6% (CR = 1/28) of the population of Beauce carry an ultra-rare pathogenic variant in CEACAM16 associated with an autosomal dominant form of deafness. Even if genetic testing in the region is limited, a marked increase in the prevalence of these clinical outcomes should be well-documented.

4 – The WGS data were taken from two separate cohorts and jointly genotyped. However technological differences can induce false positive variant calls that appear like rare variants. The authors should provide details of the quality control measures they applied to the WGS data to ensure the resulting rare variants are not technological artefacts. Currently no quality control steps for the WGS data are described. Separately, the authors could also confirm that no clustering in the 10 PCs is observed due to the two cohorts, which would likely indicate batch effects.

Minor comments:

1 – It would be helpful to see a supplementary table of the counts of individuals broken down by recruitment area/birthplace and cohort. Additionally, counts of individuals with genotype vs WGS data could also be included.

2 – The authors use bootstrapping to generate ranges of the ADR for the Montreal and SLSJ genealogical data as these groups are larger than the Beauce group. For completeness, bootstrapping could also be performed on the Beauce cohort for Figure 4. Since bootstrapping involves resampling with replacement, these ADR values for Beauce will not be constant per decade. This range should be narrow when resampling the entire group, so the interpretation of results will not change.

3 – For Figure 3, the authors could also perform bootstrapping on the mean kinship and inbreeding coefficients, for the same reason as the ADR in Figure 4.

4 – The authors use a relative frequency difference of at least 10% to identify rare variants. However, the cohort sizes are almost an order of magnitude smaller than those discussed in the related manuscript where they derive this cutoff (Michel et al., *medRxiv*, 2024). Since the allele frequency estimate will have wider confidence intervals in smaller cohorts, the cutoff of 10% may be a conservative choice for this study. Relaxing this value to account for the smaller cohort sizes might capture additional variants of interest.

5 – I find the carrier rates a little difficult to interpret when presented next to the gnomAD MAF in the tables. These could be converted to frequencies for readability, although this is a style choice.

Version 1:

Reviewer comments:

Reviewer #1

(Remarks to the Author)

Overall I appreciate the authors' effort on improving the methodology and the details for testing confounding factors like batch effect and population structure. However, the claim of a causal relationship between the identified rare variants and the founder effect might have warranted a more careful wording.

As the authors wrote in their responses, there is currently no universally accepted definition of a "founder variant" in the

literature. It means that the current methodology of identifying founder variant is quite exploratory in the field. I appreciate the novelty of this study design (as I already commented in my previously submitted), but remain skeptical about the robustness of the identified rare pathogenic variants being attributed to a founder effect.

For moving forward to the next stage of this manuscript, I would request the authors tone down the causal relation between founder effect and the identified pathogenic variant, and present the results in more careful words. As the authors explained the rebuttal letter "A high proportion of pairs of carriers sharing the same IBD segment at the variant's location indicates that it was likely inherited from a common ancestor and subsequently increased in frequency due to genetic drift, consistent with a founder effect and/or a causal relationship with the population structure." The previous wording sounds more cautious to me. These are my thoughts for your reference.

Reviewer #2

(Remarks to the Author)

The authors did a great job with this revision process and improved the article greatly. The authors effectively fixed all of my major & minor comments, including improving the bioinformatics and methodology to make this a great article. I do not have any other suggestions.

Reviewer #3

(Remarks to the Author)

I would like to commend the authors for the level of additional detail they have provided in all of their revisions, which has improved the overall quality of their manuscript. The authors have satisfactorily addressed the majority of my original concerns, and I thank them for the diligence they showed in their replies. However, I do have one outstanding issue, although I believe this will not substantively affect the results or the discussion points and should be straightforward to address.

In response to my "Major Comment 3", the authors evaluated the difference in MAF and CR for each of the founder variants between the entire cluster and its unrelated subset. However, given the sample sizes, Fisher's exact test is not sufficiently powered to identify such differences between the two groups. For example, using the "power.fisher.test" function from the "statmod" package in R, the power to detect a difference in MAF for the NM_000022.4:c.956_960del variant is 0.257 for a one-sided test assuming a sample size of 2,000 at $\alpha = 0.05$. The authors have introduced appropriate filtering steps to independently ensure that relatedness has not biased the selection of the founder variants. However, in the MAF and CR values in the main manuscript should be reported for the unrelated subset, as they cannot ensure that relatedness has not inflated the current values.

General response

We thank all the reviewers for their time and insightful comments, which have significantly contributed to improving our manuscript.

In response, we have incorporated additional details regarding the statistical and bioinformatic approaches. We have added information on the genotyping and whole-genome sequencing data, including the quality control procedures, in the supplementary material. We have also clarified the criteria used for selecting ClinVar variants. We also now provide a GitHub repository with all custom code necessary for the analyses.

Following the reviewers' suggestions, we now report confidence intervals for both the inbreeding coefficients and kinship estimates. Additionally, we revised the ADR analysis by replacing the previous minimum and maximum values with 95% confidence intervals. We also estimated the effective population size using IBD segments as proposed by reviewer #2 and compared these results with those from the genealogical analyses.

To address the reviewers' concerns about the presence of related individuals in the data, we adapted our methodology to include the related individuals while avoiding biases due to the relatedness in the cohort. Regarding the clustering analysis, we re-performed the procedure using a projection-based PCA method to mitigate bias introduced by related individuals. This resulted in a redefinition of the clusters. While there is substantial overlap with the original clusters, the revised analysis yielded an increased number of individuals assigned to the Beauce cluster and a reduced number in the UrbanQc cluster.

Using those new clusters, we reproduced our founder variant identification methodology and 24 of the 28 previously identified variants remain classified as founder variants. We also detected new founder variants that were previously undetected, while a few others no longer met the criteria. Among the four variants no longer classified as founder variants, three did not meet the threshold of five unrelated carriers (up to the third degree), and one failed to reach the minimum 50% IBD sharing at the variant position.

To ensure that the MAF and the carrier rates were not biased by the relatedness in the clusters, we calculated them in both a maximally unrelated subset and the whole cluster and ensured that there was no significant difference between the values using a Fisher exact test. Since there were no significant differences, we chose to use the whole cluster instead of filtering for relatedness to maintain a larger sample size.

In the following section, we provide a point-by-point response to every concern raised by both reviewers.

Thanks to the reviewers' constructive feedback, we refined our methodology to reduce confounding factors. We now resubmit a manuscript with a stronger methodology. But despite these various methodological updates, the main conclusions of our study remain unchanged.

Point-by-point response

Reviewer #1 (Remarks to the Author): (Note that we splitted the main comment to be clearer in our responses)

The authors present an interesting example of identifying potential founder rare pathogenic variants in a region by combining genealogical information and population structure. For population structure, the authors used UMAP. For rare variants, the authors used information from the ClinVar database.

We thank the reviewer for their appreciation.

Overall, I think the novel findings of this study should be better highlighted and the narrative could be improved. For example, the rationale for selecting rare variants is not clear and the correlation between the variants identified and their genetic background is not statistically clear.

The justification to study rare variants has been clarified in the introduction (lines 46-48; 66-69) and the selection criteria for the rare variants have been added in section 2.5 (lines 165-170).

The classification of rare pathogenic variants as more frequent in Beauce does not necessarily suggest a causal relationship between regional population structure and regional rare pathogenic variants.

While it is true that a higher frequency of a rare pathogenic variant in a specific region does not, on its own, imply a causal relationship between regional population structure and regional rare pathogenic variants, it is important to note that there is currently no universally accepted definition of a founder variant in the literature. Commonly, a variant is considered to be associated with a founder effect when it is observed at elevated frequency within a genetically related group and can be traced to one or more common ancestors¹⁻³.

In our study, we employed a two-step approach to define founder variants. We first identified variants that showed a relative frequency difference of at least 0.1 in Beauce compared to the more urban and admixed regions of Quebec. Second, we assessed IBD sharing at the variant locus among carriers. Variants for which at least 50% of carrier pairs shared an IBD segment at the variant's position were classified as founder variants. A high proportion of pairs of carriers sharing the same IBD segment at the variant's location indicates that it was likely inherited from a common ancestor and subsequently increased in frequency due to genetic drift, consistent with a founder effect and/or a causal relationship with the population structure. We made several modifications to the methodology and to the results sections to better highlight our two-step approach

The population structure could have been investigated in a more sophisticated way (e.g. by identifying ancestry sources, quantifying admixture proportions, etc.) in order to explain the heterogeneous population structure in UMAP with further analyses. Nevertheless, this study is a nice attempt to correlate pathogenic variants with population structure (and genealogy).

We agree that a more detailed characterization of the population structure, such as the identification of ancestry source or the estimation of admixture proportions could have provided additional insights into the heterogeneous patterns observed in the UMAP. However, the primary objective of this study was not to comprehensively characterize the genetic structure of the population as a whole, but rather to focus specifically on individuals of Beauce ancestry and assess the regional heightened frequency of rare pathogenic variants.

To this end, we employed UMAP and clustering as a targeted and efficient strategy to identify individuals with shared Beauce ancestry. While we acknowledge that more sophisticated methods exist, we deliberately selected this approach for its demonstrated ability to reveal fine-scale population structure while preserving broader genetic relationships⁴⁻⁷. This method effectively delineated regional clusters, aligning with our goal of studying founder effects and variant enrichment specific to the Beauce region.

Moreover, extensive analyses of Quebec's broader population structure have already been performed in previous studies, including a complete description of the genetic structure of the CARTaGENE sample used in this study⁸. In contrast, our study aimed to contribute novel insights by highlighting the relationship between regional population history and the distribution of rare pathogenic variants in a specific, under-characterized region. Therefore, we chose to prioritize this regional focus and the identification of clinically relevant variants rather than replicate population structure analyses already well-covered in the literature.

Reviewer #2 (Remarks to the Author):

Review:

I had the pleasure of reading the article "Rare variants and founder effect in an understudied Quebec population" by Gagnon et al. The authors present a compelling motivation to study the founder genealogies of Beauce and Saguenay–Lac-St-Jean (SLSJ) populations (Quebec, Canada) to understand the prevalence of pathogenic rare variants. Overall, I found the article to be well written and includes a wealth of cited sources making the article flow logically and allow the authors to test their hypotheses in a thoughtful approach.

The authors use array genotyping, whole-genome sequencing, and genealogy datasets to assess specific Quebec communities for the prevalence of pathogenic variants as well as historic estimates of ancestral diversity. I think that the authors approach is sound and clever to use such complimentary datasets and techniques. I present some issues that the authors could mitigate below and some suggested analyses that could improve the scope of the paper, although not critical for publication. I think there needs to be some additional clarity over the implementation of their pathogenic variants and the specific hypotheses this line of questions entail. Overall, the authors' findings are useful to the broader scientific community, and I think that the study provides foundational clinical evidence for their study populations in the future. I think generally, founder effects are largely forgotten to affect human populations, and it was nice to see this article address the medical consequences of small founding effective population sizes.

We sincerely thank the reviewer for their thoughtful and encouraging feedback.

Line 108-109: Why not calculate 95% confidence intervals or quantiles?

We initially wanted to show a more exhaustive portrayal of the population by showing the complete interval of ADR (minimum, mean and maximum) instead of CI or quantiles, but ultimately added 95% confidence intervals to the kinship and inbreeding so we did the same for the ADR for consistency.

Lines 113-115: I think the inclusion of familial cases of schizophrenia and bipolar disorder will skew the frequency of potentially pathogenic variants. This dataset could be used as an advantageous position to assess this conclusion. Do you prioritize these variants in any way downstream? It is not clear to me.

We appreciate the reviewer's comment.

To ensure that the inclusion of familial cases did not bias our estimates of minor allele frequencies or carrier rates, we re-calculated these metrics on a maximally unrelated subset of samples (up to the third degree). We then performed Fisher's exact test to compare these values with those obtained from the entire cluster. The results confirmed that there was no statistically significant difference between the two groups, thereby validating our approach. These details have been incorporated in Section 2.5 (lines 184-189).

We did not prioritize variants associated with schizophrenia or bipolar disorder in our analysis. Rather, we included all pathogenic rare variants listed in ClinVar, which naturally encompassed variants associated with these disorders, following certain selection criteria. We added the details about the selection in Section 2.5 (lines 165-170).

While the presence of familial cases in schizophrenia and bipolar disorder could be leveraged to explore rare variants associated with these conditions, our current study was designed to explore rare variants associated with these conditions, our current study was designed to characterize the regional population and document broader patterns of genetic variation. In this study, we only used a subset of the sample to better understand the genetic of the Beauce population, but other studies are currently focusing on the whole SZ-BP cohort to explore the potential clinical relevance of these familial cases.

Line 122-125: I think it is important to include more details regarding the bioinformatics. You have multiple Illumina technologies and the variant calling/downstream analyses could be affected by this fact. Please expand the details and make sure it is the same human genome used for mapping, if not you will have to LiftOver positions.

We thank the reviewer for their recommendation. A more detailed description of the quality control has been provided in the supplementary material for the genotyping and whole genome sequencing data.

Line 142: There is a package in R to determine the "elbow" of a scree plot that maximizes the information gained from the inclusion of multiple PCs. This might be interesting to test for your data: <https://github.com/heatherjzhou/PCAFForQTL>, <https://doi.org/10.1186/s13059-022-02761-4>

We appreciate the reviewer's recommendation. We added the elbows identified using the change point detection method (R library changepoint (version 2.2.4)) on our scree plot (supplementary figure S2).

Line 149-152: This kind of seems arbitrary to me ("The size of the epsilon neighborhood was set to 0.30, and the minimum number of points in the neighborhood was set to 30. Those parameters were chosen to ensure the identification of meaningful clusters while minimizing the inclusion of noise.").

We selected DBSCAN as the clustering method following UMAP due to its effectiveness in identifying clusters in high-dimensional spatial data without requiring a predefined number of clusters (Healy & McInnes 2024; Ester et al. 1996).

The choice of DBSCAN parameters was made after several iterations with different combinations. An epsilon neighborhood of 0.30 and a minimum number of points set to 50 were selected because they produced clusters that corresponded well with the patterns observed visually in the UMAP projection. To ensure the relevance and robustness of the identified clusters, we also validated them by comparing cluster membership with the individuals' recruitment regions. This alignment between genetic clustering and known regional origins supported the meaningfulness of the chosen

parameters. 93% of all individuals recruited in Beauce are present in the Beauce cluster. We added these precisions in section 2.4 (lines 148 and 154-155)

Line 172-173: Briefly describe how you are prioritizing ClinVar variant screening. I think this is a sticking point for me throughout the article.

As advised, precisions regarding the selection of the ClinVar variants have been added in section 2.5 (lines 165-170).

Line 184: Why non-Finnish? My gut instinct would be to not exclude any populations that make up the European dataset from gnomAD. Some justification would be helpful.

The decision to focus on non-Finnish Europeans was based on the known genetic differences between the Finnish population and other European groups. The Finnish population exhibits a distinct genetic profile due to the population bottleneck and founder effect which led to an overrepresentation of certain rare variants and the absence of some others^{9,10}. By excluding the Finnish population, we ensured that the reference MAF values reflected a more typical European population. In fact, this distinction is so widely recognised that gnomAD provides separate categories for “Europeans (Finnish) and (Europeans (non-Finnish)). While this may not be immediately obvious to all readers, it is a commonly accepted approach in genetic studies involving European populations¹¹⁻¹³.

Line 191-193: I wanted to see the inclusion of Ne estimates for your populations of interest. I know these methods are really for long-term evolutionary processes, but I think they can estimate accurately up until 1600-1700s. I saw this tool that uses IBD to estimate Ne and outputs confidence intervals, your work mentions merge-ibd-gaps as a processing step and this software has it built in: <https://faculty.washington.edu/browning/ibdne.html>; <https://doi.org/10.1371/journal.pgen.1007385>

Results 3.1: I think it would be super interesting to include some estimates of the effective population size (Ne) of the populations in Beauce and SLSJ in a proper way. I have used the MSMC2 tool in the past that can work with VCFs in your dataset following some additional pruning but it seems highly relevant for your purposes, this could also allow you to make a statement regarding the change in historic population size relative to Montreal.

We thank the reviewers for their suggestion. We estimated Ne using ibd-ne as advised and added it as Supplementary Figure S6.

Line 193: “enriched” will need to be defined, are these SNPs within windows that have an odds ratio > 3?

We thank the reviewer for noticing. This has been modified for “variant with a higher frequency in Beauce”, it was a mistake on our part.

Figure 2: This figure is a bit blurry on my end. I think it would be helpful to show how many variants / samples were included in each step. It is a bit verbose currently though.

We finally decided to exclude this figure as it did not add anything that was not already specified in the text.

Figure 3: I was confused why there are values for Beauce and SLSJ prior to the decade beginning settlement. For SLSJ, there is a stark increase in relatedness prior to settlement and it staggers off post-settlement. While Beauce shows an increase post-settlement, implying more inbreeding in Beauce than SLSJ and generally lower N_e .

Most of the initial settlers of Beauce and SLSJ were already present in the Province before the opening of those regions. For SLSJ, the post-settlement increases are more marked since approximately 80% of the initial settlers were coming from the neighboring region of Charlevoix which has been open since the end of the 17th century.

Line 261: Some of your carrier rates are below 1/20, which would equate to a frequency of less than 5%. I am confused how this works given that there are 223 individuals in your WGS dataset (for Beauce). Also, line 180 says that "Only variants present in at least five carriers and reaching a relative frequency difference of at least 10% in the Beauce cluster compared to the UrbanQc cluster were considered". So, I see a disconnect on how carriers were identified and I think it would help readers to make it more specific and add a section in the methods for this explicitly.

Does the 1/14 carrier rate mean that there is only 1 individual that is heterozygous for that variant? What does the denominator and numerator represent?

The carrier rate refers to the proportion of individuals who are heterozygous for a given variant in the population. For example, if the carrier rate is reported as 1/14, this means that 1 out of 14 individuals in the Beauce cluster is heterozygous for the variant.

As for the five carriers threshold, it was set to ensure that only variants with a sufficient representation were considered in the analysis. This threshold is based on the assumption that variants with fewer than five carriers may not provide reliable frequency estimates (A clarification was added in section 2.5, lines 178-179). When the threshold is met, the carrier rate is calculated by dividing the number of carriers (heterozygous individuals) by the number of individuals in the relevant cluster (Section 2.5 lines 183-185).

SuppTable 1 & Table 3 & SuppData1: Please give chromosome + position on HG38. Can you supply the relative frequency difference? I think it would be helpful to know the relative frequency difference for each variant as an additional column. For table 3, there is no variation in inheritance so you can remove that column and state it in the legend.

We thank the reviewer for those suggestions. The physical positions as well as the relative frequency differences have been added in the supplementary data 1. We also removed the inheritance from table 3 as advised. As for supplementary table 1, we decided to remove the table to integrate it with the Supplementary Data 1. This allowed us to add all the suggested columns.

SuppFig 7: Can you remind briefly what is the difference between founder and multiple introductions in the legend? Also, it is a bit hard to see the shapes at the bottom, you could think about using `facet_wrap(~Variant classification)` to split the plot (I am assuming you are using ggplot2 in R) into the two categories instead.

We thank the reviewers for this suggestion, we applied the modification and briefly reminded the definition of founder and multiple introduction variants in the legend.

General comments on bioinformatics: I admire that you were rigorous with showing which versions were used for all software.

Data availability: You need to include a GitHub or similar coding repository for all of the code and scripts for your project.

We thank the reviewer for their appreciation, as for the data availability, we followed the recommendation and added a GitHub repository with all the original code necessary for the analyses.

Reviewer #3 (Remarks to the Author):

Overview:

Gagnon and colleagues present a population-level genomic analysis of different regions in the Quebec province. Using genealogical data, the authors show clear evidence of founder effects in Beauce and Saguenay-Lac-St-Jean, distinct from the more ancestrally diverse area of Montreal. The authors describe the history and dynamics of settlement in the area and how that aligns with the observed genealogical data. Next, the authors use an identity-by-descent approach to identify clinically relevant variants in Beauce likely arising from founders. This work is in general well conceptualised and represents the first exploratory analysis of the clinical genomic landscape of the Beauce region. However, despite the intriguing results, I have some issues with the methodology which may impact the conclusions of this manuscript.

We thank the reviewer for their appreciation and their suggestions.

Major comments:

1 – In Section 2.4, the authors are rightly concerned about how PC shrinkage following a projection may cloud the fine population structure. However, their decision not to adjust for the relatedness in the cohort will have impacts downstream that cannot be alleviated. For example, the majority of the Beauce cluster from the UMAP analysis is derived from the SZ-BD family cohort. The authors cannot know if this distinct clustering is a result of population structure in Beauce or due the high degree of relatedness in the selected individuals (likely both). To avoid this issue, the authors could either take a maximally unrelated subset of the SZ-BD cohort or adjust for the relatedness in the PCA via projection. Indeed, the publication the authors cite in regard to this discusses strategies to reduce PC shrinkage following projection (Privé et al., *Bioinformatics*, 2020). Regardless, the relatedness as described should not be left unaccounted for.

We thank the reviewer for their suggestion. To ensure that our clusters were indeed representative of the population structure and not biased by the presence of related individuals, we modified our approach and selected a subset of unrelated individuals (up to the third degree) and projected the related individuals using the PC-AiR function from the GENESIS library. The methodology in section 2.4 has been updated accordingly. This still allowed the identification of regional clusters with a great overlap with the previous clusters. This allowed us to include more individuals in the Beauce cluster and led to the formation of a smaller cluster associated with the urban regions. 44% of the individuals in the Beauce cluster have been recruited in Beauce compared to the previous 65%, but 93% of all individuals recruited in Beauce are comprised in the cluster. Because of the inclusion of more individuals, there are some differences in the carrier rates, but we retained almost all previously identified founder variants and were able to detect some more.

2 – I am not convinced that it is appropriate to treat the clinically-derived SZ-BD cohort as a population-representative group. In general, individuals with schizophrenia or bipolar disorder likely represent less than 3-4% of the general population and are enriched for rare, damaging variants. As

such, it may not be surprising to see rare pathogenic variants in these individuals or their close relatives. If this cohort is to be retained, the authors should detail their attempts to mitigate the selection bias compared to the more broadly recruited CaG cohort. Otherwise, the results described may not generalise to the entire Beauce region.

To our knowledge, there is no evidence of an enrichment of rare variants associated with general conditions in individuals with SZ or BD. Furthermore, the selected ClinVar variants include rare variants associated with schizophrenia and bipolar disorders and none were detected as more frequent in the Beauce cluster compared to the UrbanQc cluster.

To make sure there was not an elevated burden in the affected individuals, we first checked the diagnosis of individuals carrying one or more variants with a higher frequency in Beauce. We excluded individuals that were never assessed by a psychiatrist to ensure there was no bias caused by an unknown diagnosis and therefore only kept affected and unaffected individuals. A similar proportion of affected or unaffected individuals carry the same amount of variants and no significant differences were detected between the values as assessed by a chi-square test. These results suggest that the observed variant burden is not driven by disease status within the SZ-BD cohort.

Chi-square test between the affected and unaffected carriers for each number of carried variant with a higher frequency in Beauce

Number of variants	Proportion of affected individuals	Proportion of unaffected individuals	P-Value
1	20/60	43/125	1.0000
2	16/60	44/125	0.3208
3	14/60	27/125	0.9389
4	6/60	7/125	0.4302
5	1/60	2/125	1.0000
6	3/60	0/125	0.0576
7	0/60	2/125	0.8214

We also verified founder variants individually to ensure that they were present in patients as well as non-affected individuals. While some variants are never seen among patients, all variants are carried by unaffected individuals. We performed a chi-square test to verify if the difference between the number of carrier were statistically significant and it was only the case for variants c.2842del which is only seen amongst unaffected individuals.

Chi-square test between the affected and unaffected carriers for each founder variant.

HGVS	Number of affected carriers	Number of unaffected carriers	P-value
c.1001+3_1001+6del	7	8	0.79625341
c.1054C>T	3	6	0.31731051
c.1100del	3	2	0.65472085
c.1103G>A	1	5	0.10247043
c.113G>A	2	2	1

c.1171A>G	0	3	0.08326452
c.1186A>G	2	6	0.15729921
c.1274_1277dup	3	3	1
c.1474C>T	1	5	0.10247043
c.1510C>T	1	6	0.05878172
c.15918-12A>G	3	7	0.20590321
c.2094C>G	3	3	1
c.2275C>T	3	4	0.70545699
c.2436+1del	3	3	1
c.262del	3	2	0.65472085
c.2842del	0	5	0.02534732
c.3113C>T	2	6	0.15729921
c.370C>T	5	12	0.08955507
c.3749T>C	1	4	0.17971249
c.413G>A	0	3	0.08326452
c.529G>A	2	6	0.15729921
c.610-2A>G	3	2	0.65472085
c.617T>G	2	4	0.41421618
c.635T>C	4	3	0.70545699
c.656-2_656-1del	7	12	0.25134911
c.6598A>T	2	6	0.15729921
c.6724C>T	5	4	0.73888268
c.6793C>T	2	7	0.0955807
c.67A>G	5	10	0.1967056
c.746_749del	2	3	0.65472085
c.766C>T	1	4	0.17971249
c.767_786del	3	2	0.65472085
c.856C>T	5	9	0.28504941
c.8606C>G	3	4	0.70545699
c.932C>T	5	8	0.40538056
c.956_960del	7	11	0.34577859

3 – In Sections 2.5 and 3.3, MAFs and CRs for rare variants of interest are described to prioritise those found more frequently in Beauce. The authors should confirm that these values were calculated on an unrelated subset of the respective groups, since the high degree of relatedness from SZ-BD will incorrectly inflate the MAFs. Perhaps I am misinterpreting, but some of the pathogenic variants listed in Supplementary Table 1 are quite common in the Beauce cohort (equivalent MAF of 2-7%) but are extremely rare or absent in the well-powered gnomAD database. It doesn't seem plausible to me that, for example, ~3.6% (CR = 1/28) of the population of Beauce carry an ultra-rare pathogenic variant in CEACAM16 associated with an autosomal dominant form of deafness. Even if genetic testing in the region is limited, a marked increase in the prevalence of these clinical outcomes should be well-documented.

We appreciate the reviewer's comment regarding the potential inflation of MAF and carrier rate (CR) estimates due to relatedness within the SZ-BP cohort. The concern is valid, particularly in the context of rare variants where familial aggregation can disproportionately affect frequency estimates. In order to account for the relatedness, we selected a maximally unrelated subset (up to the third degree) of each cluster and calculated the MAF and CR for each founder variant. We then compared them to the MAF and CR in the whole cluster, which includes related individuals. To assess whether relatedness introduced a significant bias in allele frequency estimates, we performed Fisher's exact tests comparing allele counts and carrier counts between the unrelated subset and the full cluster. For all founder variants, these tests returned non-significant p-values ($p > 0.05$), indicating that the presence of related individuals did not significantly increase the MAF or CR estimates. MAF were slightly higher in the whole cluster than in the unrelated sample for only four founder variants and the difference was not statistically significant. Therefore, we chose to retain the values calculated on the whole sample instead of the unrelated subset to maximize our sample size.

In addition, to minimize the likelihood that an apparent elevated frequency in Beauce was driven by a single family or a small number of related carriers, we applied a carrier count filter, retaining only variants found in at least five unrelated individuals. This ensured that the observed elevated frequencies reflected broader regional prevalences rather than familial aggregation.

Together, these measures support the robustness of our reported MAF and CR estimates and confirm that they are not substantially confounded by relatedness within the SZ-BP cohort.

As for the variant NM_001039213.4:c.1186A>G on the CEACAM16 gene, the CR we calculated in the new Beauce cluster (formed using a projection of the related individuals on the unrelated population) is now slightly lower than what we previously estimated (1/40 instead of 1/28). When calculated on the unrelated subset, the CR rises to 1/29 because of the smaller sample size, but this difference remains insignificant. Also, further investigation made us realise that this variant has been observed to cause disease under different inheritance patterns (autosomal dominant or autosomal recessive) that might help explain why there hasn't been any evidence of increased deafness in Beauce so far. This clarification has been added in the Supplementary Data 1 and the discussion was updated accordingly (lines 394-404).

4 – The WGS data were taken from two separate cohorts and jointly genotyped. However technological differences can induce false positive variant calls that appear like rare variants. The authors should provide details of the quality control measures they applied to the WGS data to ensure the resulting rare variants are not technological artefacts. Currently no quality control steps for the WGS data are described. Separately, the authors could also confirm that no clustering in the 10 PCs is observed due to the two cohorts, which would likely indicate batch effects.

As advised, we added more details about the genetic data and quality control (see supplementary material). As for the batch effect, we compared the MAF of the ClinVar rare pathogenic variants between the two cohorts to ensure that there were no batch effect (Supplementary Figure S5). We also ensured that there were individuals from both cohorts in the Beauce and UrbanQc cluster and that we observed founder variants in both (Supplementary Figure S8). Despite the fact that all individuals recruited in Beauce were from the SZ-BP cohort, some individuals from CaG still clustered with Beauce individuals due to their ancestry. While not included in the article, we provide the reviewer with the PCA figure colored by cohort to show that there was no batch effect and that the PCA represents the difference between regional ancestries and not between cohorts.

Principal component analysis of the genotype data with individuals colored by cohort

Minor comments:

1 – It would be helpful to see a supplementary table of the counts of individuals broken down by recruitment area/birthplace and cohort. Additionally, counts of individuals with genotype vs WGS data could also be included.

We chose not to include a breakdown per recruitment region as we considered the ancestry of the cluster over the recruitment region and did not want to confuse the readers. We believe that supplementary figure S4 would be sufficient to illustrate the recruitment region of the clustered individuals. To accompany the count of genotypes vs WGS (Table 2), we added the precision that all sequenced individuals were previously genotyped (section 2.5 line 163).

2 – The authors use bootstrapping to generate ranges of the ADR for the Montreal and SLSJ genealogical data as these groups are larger than the Beauce group. For completeness, bootstrapping

could also be performed on the Beauce cohort for Figure 4. Since bootstrapping involves resampling with replacement, these ADR values for Beauce will not be constant per decade. This range should be narrow when resampling the entire group, so the interpretation of results will not change.

We thank the reviewer for their advice, we did not allow resampling with replacement at submission time, so we remade the analyses to adjust it accordingly. We also added 95% confidence intervals instead of the maximal and minimal values to remain consistent.

3 – For Figure 3, the authors could also perform bootstrapping on the mean kinship and inbreeding coefficients, for the same reason as the ADR in Figure 4.

Since average kinship and inbreeding are less influenced by the sample size than the ADR, we chose to include the whole sample. However, we did add 95% confidence intervals based on 1,000 bootstrapped simulations.

4 – The authors use a relative frequency difference of at least 10% to identify rare variants. However, the cohort sizes are almost an order of magnitude smaller than those discussed in the related manuscript where they derive this cutoff (Michel et al., medRxiv, 2024). Since the allele frequency estimate will have wider confidence intervals in smaller cohorts, the cutoff of 10% may be a conservative choice for this study. Relaxing this value to account for the smaller cohort sizes might capture additional variants of interest.

Given the limited literature related to the regional variants with a higher frequency in Beauce, we preferred using a more stringent cutoff point and identifying fewer variants than risk having false positives. We preferred to remain conservative, especially because of concerns regarding the size of our sample and about the generalization of our results to the general Beauce population. We acknowledge that this conservative threshold may have limited the number of variants detected and possibly excluded variants of potential interest. Access to clinical data and phenotypes would have been an asset to allow a more relaxed threshold while offering robust conclusions. This limitation has been added to the manuscript (lines 181-183; 412-414)

5 – I find the carrier rates a little difficult to interpret when presented next to the gnomAD MAF in the tables. These could be converted to frequencies for readability, although this is a style choice.

To facilitate the interpretation, MAF for the Beauce and UrbanQc clusters have been added to the Supplementary Data 1.

References :

1. Jain, A., Sharma, D., Bajaj, A., Gupta, V. & Scaria, V. Chapter Four - Founder variants and population genomes—Toward precision medicine. in *Advances in Genetics* (ed. Kumar, D.) vol. 107 121–152 (Academic Press, 2021).
2. Isshiki, M. *et al.* Genetic disease risks of under-represented founder populations in New York City. 2024.09.27.24314513 Preprint at <https://doi.org/10.1101/2024.09.27.24314513> (2024).
3. Marafi, D. Founder mutations and rare disease in the Arab world. *Dis. Model. Mech.* **17**, (2024).

4. Diaz-Papkovich, A., Anderson-Trocmé, L., Ben-Eghan, C. & Gravel, S. UMAP reveals cryptic population structure and phenotype heterogeneity in large genomic cohorts. *PLoS Genet.* **15**, e1008432 (2019).
5. Diaz-Papkovich, A., Anderson-Trocmé, L. & Gravel, S. A review of UMAP in population genetics. *J. Hum. Genet.* **66**, 85–91 (2021).
6. Healy, J. & McInnes, L. Uniform manifold approximation and projection. *Nat. Rev. Methods Primer* **4**, 82 (2024).
7. Diaz-Papkovich, A. *et al.* Topological stratification of continuous genetic variation in large biobanks. 2023.07.06.548007 Preprint at <https://doi.org/10.1101/2023.07.06.548007> (2023).
8. McClelland, P. *et al.* A multi-ancestry genetic reference for the Quebec population. 2025.05.14.25327536 Preprint at <https://doi.org/10.1101/2025.05.14.25327536> (2025).
9. Uusimaa, J. *et al.* The Finnish genetic heritage in 2022 – from diagnosis to translational research. *Dis. Model. Mech.* **15**, dmm049490 (2022).
10. Kere, J. Human Population Genetics: Lessons from Finland. *Annu. Rev. Genomics Hum. Genet.* **2**, 103–128 (2001).
11. Kandolin, M., Pöyhönen, M. & Jakkula, E. Estimation of carrier frequencies utilizing the gnomAD database for ACMG recommended carrier screening and Finnish disease heritage conditions in non-Finnish European, Finnish, and Ashkenazi Jewish populations. *Am. J. Med. Genet. A.* **194**, e63588 (2024).
12. Nappo, S. *et al.* Carrier frequency of CFTR variants in the non-Caucasian populations by genome aggregation database (gnomAD)-based analysis. *Ann. Hum. Genet.* **84**, 463–468 (2020).
13. Michel, É. *et al.* Rare diseases load through the study of a regional population. *medRxiv* 2024.10.29.24316346 (2024).

We thank the reviewers for their appreciation of our revised manuscript. Following their suggestions, we addressed the clarity in wording our findings and modified the carrier rates and MAF in the main text to report only the values calculated on the unrelated subset. Those slight modifications, in addition to the one we already made for the previous submission, help improve the clarity and robustness of our manuscript.

In response to Reviewer #1, we carefully revised the wording of our claims regarding the identification of founder variants to avoid implying a direct causal relationship between the heightened frequency of the variant and the founder event.

In response to Reviewer #3, we acknowledge the limited power of Fisher's exact test given the sample sizes and have followed the reviewer's recommendation to report MAF and carrier rates calculated on the maximally unrelated subset in the main text.

Reviewer #1 (Remarks to the Author):

Overall I appreciate the authors' effort on improving the methodology and the details for testing confounding factors like batch effect and population structure. However, the claim of a causal relationship between the identified rare variants and the founder effect might have warranted a more careful wording.

As the authors wrote in their responses, there is currently no universally accepted definition of a "founder variant" in the literature. It means that the current methodology of identifying founder variant is quite exploratory in the field.

I appreciate the novelty of this study design (as I already commented in my previously submitted), but remain skeptical about the robustness of the identified rare pathogenic variants being attributed to a founder effect.

For moving forward to the next stage of this manuscript, I would request the authors tone down the causal relation between founder effect and the identified pathogenic variant, and present the results in more careful words. As the authors explained the rebuttal letter "A high proportion of pairs of carriers sharing the same IBD segment at the variant's location indicates that it was likely inherited from a common ancestor and subsequently increased in frequency due to genetic drift, consistent with a founder effect and/or a causal relationship with the population structure." The previous wording sounds more cautious to me. These are my thoughts for your reference.

We thank the reviewer for their appreciation and suggestion of improvement. In the revised manuscript, we have carefully revised the language throughout the text to avoid implying a direct causal relationship between the elevated frequency of a variant and the founder effect. Instead, we added the precisions that the elevated frequency and the IBD sharing between the carriers are indicative of an introduction by a common ancestor followed by a spread by drift in the population. We now emphasize that the results are consistent with a founder effect, without overstating the causal inference. These changes have been made throughout the manuscript, as reflected in the tracked revisions.

Reviewer #2 (Remarks to the Author):

The authors did a great job with this revision process and improved the article greatly. The authors effectively fixed all of my major & minor comments, including improving the bioinformatics and methodology to make this a great article. I do not have any other suggestions.

We thank the reviewer for their positive feedback and appreciation of our revisions. We are pleased that the improvements to the methodology and clarity of presentation have strengthened the manuscript.

Reviewer #3 (Remarks to the Author):

I would like to commend the authors for the level of additional detail they have provided in all of their revisions, which has improved the overall quality of their manuscript. The authors have satisfactorily addressed the majority of my original concerns, and I thank them for the diligence they showed in their replies. However, I do have one outstanding issue, although I believe this will not substantively affect the results or the discussion points and should be straightforward to address.

In response to my “Major Comment 3”, the authors evaluated the difference in MAF and CR for each of the founder variants between the entire cluster and its unrelated subset. However, given the sample sizes, Fisher’s exact test is not sufficiently powered to identify such differences between the two groups. For example, using the “power.fisher.test” function from the “statmod” package in R, the power to detect a difference in MAF for the NM_000022.4:c.956_960del variant is 0.257 for a one-sided test assuming a sample size of 2,000 at $\alpha = 0.05$. The authors have introduced appropriate filtering steps to independently ensure that relatedness has not biased the selection of the founder variants. However, in the MAF and CR values in the main manuscript should be reported for the unrelated subset, as they cannot ensure that relatedness has not inflated the current values.

We thank the reviewer for their encouraging remarks and recognition of the revisions made to the manuscript. We acknowledge the reviewer’s point and agree that Fisher’s exact test is underpowered in this context. We now report only the values derived from the unrelated subset in the main text and in the Supplementary Data 2 table containing all information related to the observed variants. To maintain transparency and allow readers to observe the overall MAF and carrier rates, we have retained the values from the full sample in Supplementary Data 1.